# Constrained Adjusted Maximum a Posteriori Estimation of Bayesian Network Parameters

**DOI:** 10.3390/e23101283

**Published:** 2021-09-30

**Authors:** Ruohai Di, Peng Wang, Chuchao He, Zhigao Guo

**Affiliations:** 1School of Electronics and Information Engineering, Xi’an Technological University, Xi’an 710021, China; diruohai@xatu.edu.cn (R.D.); wang_peng@xatu.edu.cn (P.W.); hechuchao@xatu.edu.cn (C.H.); 2School of Electronic Engineering and Computer Science, Queen Mary University of London, London E1 4NS, UK

**Keywords:** graphical models, domain knowledge, prior distribution, equivalent sample size, parameter constraints

## Abstract

Maximum a posteriori estimation (MAP) with Dirichlet prior has been shown to be effective in improving the parameter learning of Bayesian networks when the available data are insufficient. Given no extra domain knowledge, uniform prior is often considered for regularization. However, when the underlying parameter distribution is non-uniform or skewed, uniform prior does not work well, and a more informative prior is required. In reality, unless the domain experts are extremely unfamiliar with the network, they would be able to provide some reliable knowledge on the studied network. With that knowledge, we can automatically refine informative priors and select reasonable equivalent sample size (ESS). In this paper, considering the parameter constraints that are transformed from the domain knowledge, we propose a Constrained adjusted Maximum a Posteriori (CaMAP) estimation method, which is featured by two novel techniques. First, to draw an informative prior distribution (or prior shape), we present a novel sampling method that can construct the prior distribution from the constraints. Then, to find the optimal ESS (or prior strength), we derive constraints on the ESS from the parameter constraints and select the optimal ESS by cross-validation. Numerical experiments show that the proposed method is superior to other learning algorithms.

## 1. Introduction

A Bayesian network (BN) is a type of graphical model that combines probability and causality theory. A BN becomes a causal model that enables reasoning about intervention under a desired causal assumption [1,2,3]. BNs have been shown to be powerful tools for addressing statistical prediction and classification problems, and they have been widely applied in many fields, such as geological hazard prediction [4], reliability analysis [5,6], medical diagnosis [7,8], gene analysis [9], fault diagnosis [10], and language recognition [11]. A BN B=(G,Θ) includes two components: a graph structure G and a set of parameters Θ. The structure G is a Directed Acyclic Graph (DAG) that consists of nodes (also called vertices) representing random variables,$\;(X_{1},\ldots ,X_{n})$, where *n* is the number of variables, and directed edges (also called arcs) correspond to the conditional dependence relationships among the variables. Notice that there should be no directed cycles in the graph. When sufficient data are available, the parameters of BN can be precisely and efficiently learnt by statistical approaches such as Maximum Likelihood (ML) estimation. When the sample data set is small, ML estimation often overfits the data and fails to approximate the underlying parameter distribution. To address this problem, Maximum a Posteriori (MAP) estimation has been introduced and shown to be effective in improving parameter learning. Because of the useful properties, i.e., (I) hyper-parameters of the BN model can be taken as equivalent sample observations and (II) experts find it convenient to define the uniformity of the distribution, the Dirichlet distribution is often preferred for the discrete BN model and therefore added into the estimating process. For the sake of clarity, we define the MAP parameter estimation of node i as $\left(N_{ijk}+\alpha_{ijk}\right)/\left(N_{ij}+\alpha_{ij}\right)$. $N_{ijk}$ is the number of observations in the data set where node i has the *k*th state and its set of parents has the *j*th state of its configurations. $N_{ij}$ is the sum of $N_{ijk}$ over all *k*. $\alpha_{ijk}$ and $\alpha_{ij}$ are the equivalent numbers of $N_{ijk}$ and $N_{ij}$ in prior beliefs. For all k, $\alpha_{ijk}$ is also the hyper-parameter values of the Dirichlet prior distribution of the BN parameter $\theta_{ijk}$, and $\alpha_{ij}$ is also the prior strength or equivalent sample size (ESS).

Given no extra domain knowledge, a uniform prior or flat prior is often chosen among all the candidate Dirichlet priors. Based on the uniform prior, MAP scores, such as Bayesian Dirichlet uniform (BDu) [12], Bayesian Dirichlet equivalent uniform (BDeu) [13] and Bayesian Dirichlet sparse (BDs) [14] have been developed and investigated [15,16,17,18,19]. When the underlying parameter distribution is uniform, (I) if the distribution obtained by purely data-driven estimation $N_{ijk}/N_{ij}$ for the parameter $\theta_{ijk}\;$ is also uniform, the selection of ESS has minor effects on MAP estimation and (II) if the distribution obtained by purely data-driven estimation $N_{ijk}/N_{ij}$ for the parameter $\theta_{ijk}$ is non-uniform, the ESS becomes crucial and the MAP estimation only approximates the underlying distribution by a large ESS value. However, when the underlying parameter distribution is non-uniform, the uniform prior becomes non-informative and, no matter what size the ESS value is, the MAP estimation based on the uniform prior fails to approximate the underlying distribution. Therefore, a well-defined or informative prior is significant.

In practice, unless the domain experts are totally unfamiliar with the studied problem, they would be able to provide some prior information about the underlying parameters [20,21], e.g., parameter A is very likely to be larger than 0.6, or parameter A is larger than B. In this paper, we assume that the expert opinion or domain knowledge is trustworthy, i.e., the domain knowledge would not be incorporated into the parameter estimation unless the domain experts are confident about their opinions. In fact, this is the assumption that many existing parameter estimation algorithms rely on [22,23,24,25,26]. From the reliable domain knowledge, we can refine informative priors. Then, with an informative prior, we can further select a reasonable ESS. In view of the above considerations, we conclude that, to obtain accurate MAP estimation, informative prior distribution is required to represent the given domain knowledge and thereby select the reasonable ESS to balance the impact of data and prior. Based on such an idea, in this paper, we present a Constrained adjusted Maximum a Posteriori (CaMAP) estimation approach to learn the parameter of a discrete BN model.

This paper is organized as follows. Section 2 briefly introduces related concepts and the studied problem. Section 3 focuses on the illustration of a novel prior elicitation algorithm and a novel optimal ESS selection algorithm. Section 4 presents the experimental results of the proposed method. Finally, we summarize the main findings of the paper and briefly explore the directions for future research in Section 5.

## 2. The Background

### 2.1. Bayesian Network

A BN is a probabilistic graphical model representing a set of variables and their conditional dependencies via a DAG. Learning a BN includes two parts: structure learning and parameter learning. Structure learning consists of finding the optimal DAG *G* that identifies the dependencies between variables from the observational data. Parameter learning entails estimating the optimal parameters θ that quantitatively specify the conditional dependencies between variables. Given the structure, the parameter estimation of a network can be factorized into the independent parameter estimations of individual variables, which means:(1)$$\ell (D|\theta )=\overset{n}{\underset{i=1}{\sum }}\overset{q_{i}}{\underset{j=1}{\sum }}\overset{r_{i}}{\underset{k=1}{\sum }}N_{ijk}log\theta_{ijk}$$
where ℓ(D|θ) is the likelihood function of parameters θ given observational data D, and the ML estimation of parameter $\theta_{ijk}$ is
(2)$$\theta_{ijk}=\frac{N_{ijk}}{N_{ij}}$$
where $N_{ij}=\sum_{k=1}^{r_{i}}N_{ijk}$.

When the observed data are sufficient, the ML estimation often fits the underlying distributions well. However, when the data are insufficient, additional information such as domain knowledge is required to prevent over-fitting.

### 2.2. Parameter Constraints

Domain knowledge can be transformed into qualitative parameter constraints. In practice, there are three common parameter constraints [22,27], which are all convex (i.e., the constraints form a convex constrained parameter feasible set that is easy to compute its geometric center, see Section 3.1). The constraints are:

(1) Range constraint: This constraint defines the upper and lower bounds of a parameter, and it is commonly considered in practice.
(3)$$\theta_{ijk}^{lower}\leq \theta_{ijk}\leq \theta_{ijk}^{upper}$$

(2) Intra-distribution constraint: This constraint describes the comparative relationship between two parameters that refer to the same parent configuration state but different child node states.
(4)$$\theta_{ijk}\leq \theta_{ijk^{\prime }},\forall k\neq k^{\prime }$$

(3) Cross-distribution constraint: This constraint has also been called “order constraint” [23] or “monotonic influence constraint” [24]. It defines the comparative relationship between two parameters that share the same child node state but different parent configuration node states.
(5)$$\theta_{ijk}\leq \theta_{ij^{\prime }k},\forall j\neq j^{\prime }$$
The third type of constraints might be hard to understand. As an example, smoking (*S* = 1) and polluted air (*PA* = 1) are two causes of lung cancer (*LC* = 1) and medical experts agree that smoking is more likely to cause lung cancer. Then, the medical knowledge could be expressed as a cross-distribution constraint, P(*C* = 1|*S* = 1, *PA* = 0) > P(*C* = 1|*S* = 0, *PA* = 1).

### 2.3. Problem Formulation

With observational data and domain knowledge, the parameter learning problem of a discrete BN can be formally defined as:


**Input:**
n: Number of nodes in the network.G: Structure with unknown parameters.D: Set of complete observations for variables.Ω: Set of parameter constraints transformed from reliable domain knowledge, $\Omega =\left\{\Omega_{1},\Omega_{2},\ldots ,\Omega_{n}\right\}$, where $\Omega_{i}$ denotes all the constraints on node i.


**Task:** Find the optimal parameters that approximate the underlying parameter distribution, $\hat{\theta }=\left\{{\hat{\theta }}_{1},\;\ldots ,{\hat{\theta }}_{n}\right\}$, ${\hat{\theta }}_{i}=\left\{{\hat{\theta }}_{i1},\;\ldots ,{\hat{\theta }}_{iq_{i}}\right\}$, ${\hat{\theta }}_{ij}=\left\{{\hat{\theta }}_{ij1},\;\ldots ,{\hat{\theta }}_{ijr_{i}}\right\}$. Here, $q_{i}$ is the number of configuration state values of the parents of the variable $X_{i}$ and $r_{i}$ is the number of state values of the variable $X_{i}$.

### 2.4. Sample Complexity of BN Parameter Learning

Basically, the ML estimation method learns accurate parameters when the acquired data are sufficient. However, when the data are insufficient, ML estimation is often inaccurate. Thus, definition of sample complexity for BN parameter learning helps to determine whether ML meets the accuracy requirement. With regard to this problem, Dasgupta [28] defined the lower bound of the sample size for BN parameters learning with known structures. Given that a network has n binary variables, and no node has more than k parents, then the sample complexity with confidence 1−δ is lower bounded by
(6)$$\frac{288\times n^{2}\times 2^{k}}{\varepsilon^{2}}\times \ln^{2}\left(1+\frac{3n}{\varepsilon }\right)\times \ln\left(\frac{1+3n/\varepsilon }{\varepsilon \delta }\right)$$
where ε is the error rate and is often computed as ε=nσ, for a small constant σ.

## 3. The Method

Among all the parameter learning algorithms, MAP estimation is a learning algorithm that conveniently combines the prior knowledge and observed data. For node i, the posteriori estimation of parameters $\theta_{ij}$ can be written as
(7)$$P(\theta_{ij}|D)=\frac{P(D|\theta_{ij})P\left(\theta_{ij}\right)}{P\left(D\right)}\propto P(D|\theta_{ij})P\left(\theta_{ij}\right)$$
where $P\left(\theta_{ij}\right)$ denotes the prior distribution and $P(D|\theta_{ij})$ equals to $l(D|\theta_{ij})$. Thus, the MAP estimation of ${\hat{\theta }}_{ij}$ can be further defined as:(8)$${\hat{\theta }}_{ij}=\underset{\theta_{ij}}{argmax}P(\theta_{ij}|D)=\underset{\theta_{ij}}{argmax}P(D|\theta_{ij})P\left(\theta_{ij}\right)$$

Since the parameters $\theta_{ij}$ studied in this paper follows the multinomial distribution and the conjugate prior for the multinomial distribution is Dirichlet distribution, the prior distribution of $\theta_{ij}=\left(\theta_{ij1},\ldots ,\theta_{ijr_{i}}\right)$ is set to be the Dirichlet distribution, i.e., $\theta_{ij}\sim Dir\left(\alpha_{ij1},\ldots ,\alpha_{ijr_{i}}\right)$, where $\left(\alpha_{ij1},\ldots ,\alpha_{ijr_{i}}\right)$ are the priors equivalent to the observations $\left(N_{ij1},\ldots ,N_{ijr_{i}}\right)$. As a result, the approximate MAP estimation (see Appendix A) for $\theta_{ijk}$ has the form
(9)$${\hat{\theta }}_{ijk}=\frac{N_{ijk}+\alpha_{ijk}}{N_{ij}+\alpha_{ij}}$$
where $\alpha_{ij}=\sum_{k=1}^{r_{i}}\alpha_{ijk}$ is the equivalent (or hypothetical) sample size.

Generally, domain experts would find it difficult to provide a specific prior Dirichlet distribution but feel more comfortable to make qualitative statements on unknown parameters. From such qualitative parameter statements or parameter constraints, the prior distribution $Dir\left(\alpha_{ij1},\ldots ,\alpha_{ijr_{i}}\right)$ can be further defined as
(10)$$Dir\left(\alpha_{ij1},\ldots ,\alpha_{ijr_{i}}\right)=Dir\left(\alpha_{ij}*\theta_{ij}^{prior}\right)$$
where $\theta_{ij}^{prior}=\left(\theta_{ij1}^{prior},\theta_{ij2}^{prior},\ldots ,\theta_{ijr_{i}}^{prior}\right)$ is the prior hyper-parameter vector of the prior distribution that represents the domain knowledge and can be sampled from the parameter constraints. Finally, the MAP estimation for $\theta_{ijk}$ can be expressed as
(11)$${\hat{\theta }}_{ijk}=\frac{N_{ijk}+\alpha_{ij}\theta_{ijk}^{prior}}{N_{ij}+\alpha_{ij}}$$

As the parameter constraints are incorporated into the MAP estimation, we define the above estimation as Constrained adjusted Maximum a Posteriori (CaMAP) estimation. In the following sections, we will introduce the elicitation of the prior parameter $\theta_{ij}^{prior}$ and the selection of the optimal ESS $\alpha_{ij}$.

### 3.1. Prior Elicitation

Before defining the optimal ESS $\alpha_{ij}$, the prior parameter $\theta_{ij}^{prior}$ is required, which could be elicited from the parameter constraints in a sampling manner. In this paper, we design a sampling method that applies to all types of convex constraints. Specifically, in the sampling method,

(1) First, we search for the optimal parameters of the following model:(12)minimize C
(13)$$subject\;to\;\Omega \left(\theta_{i}\right)$$
where C is a random constant and $\Omega \left(\theta_{i}\right)$ represents all the parameter constraints on node i. The constrained model is simple and could be efficiently solved. Note that even though the objective function is a constant, the solutions of the constrained model could vary each time. In fact, any parameters satisfying the given parameter constraints are solutions of the constrained model. Therefore, through iteratively solving the constrained model, we collect the parameters that cover the feasible parameter region constrained by the parameter constraints.

(2) Then, the first step is repeated (In this paper, we set the repetition times at 100 and the sampling code is available at: https://uk.mathworks.com/matlabcentral/fileexchange/34208-uniform-distribution-over-a-convex-polytope (accessed on 26 September 2021)) to collect sufficient sampled parameters that cover the constrained parameter space. To make sure that the sampled parameters are uniformly distributed over the constrained parameter space, for each sampling step, we add an extra constraint
(14)$$\|\theta_{i}^{t+1}-\theta_{i}^{t}{\|}_{2}\geq \tau$$
where τ is a small value (e.g., 0.1), $\theta_{i}^{t}$ represents the sampled parameters at step t, and $\theta_{i}^{t+1}$ represents the sampled parameters at step t+1.

(3) Finally, we average over all the sampled parameters and set the mean values as the prior $\theta_{i}^{prior}=\left\{\theta_{ij}^{prior}\right\}$, $j=\left\{1,\ldots q_{i}\right\}$, where $\theta_{ij}^{prior}=\left(\theta_{ij1}^{prior},\ldots ,\theta_{ijr_{i}}^{prior}\right)$.

### 3.2. ESS Value Selection

Although the sampled prior $\theta_{i}^{prior}$ guarantees satisfying all the parameter constraints, the overall estimation (Equation (10)) may violate the constraints if ESS $\alpha_{ij}$ is not reasonably defined. For example, for binary variables, {LC = Lung Cancer, S = Smoking, PA= Pollution Air}, smoking and pollution air are shown to cause lung cancer. Parameter $\theta_{142}$ represents the probability that the value of variable LC is true given that the values of variables S and PA are both true. In this example, $\theta_{142}$ is the probability of having lung cancer (LC=1) given that the patients consistently smoke (S=1) and work in polluted air (PA=1). The medical experts assert that $\theta_{142}$ lies in the interval, [0.6, 1.0], which is also the parameter constraint. Now, the elicited prior $\theta_{142}^{prior}$ is 0.80, which satisfies the parameter constraint, and the purely data-driven estimation (also ML estimation) is $N_{142}/N_{14}=1/7$. Then, with a small ESS, such as 5, the estimation (Equation (11)) is computed as follows:(15)$${\hat{\theta }}_{142}=\frac{1+5*0.80}{7+5}=0.42$$

Obviously, the above estimation does not satisfy the constraint, $\theta_{142}\in \left[0.6,1.0\right]$. In fact, to make sure that the estimation does not violate the constraint, the optimal ESS should not be less than 16, which could be inferred from the parameter constraints. Therefore, given the elicited prior and observation counting, to guarantee that the overall CaMAP estimation satisfies all the parameter constraints, the optimal ESS should satisfy certain constraints.

From each type of constraint imposed on the parameters, ESS constraints could be derived as follows:

(1) To satisfy the range constraint, the CaMAP estimation in Equation (11) should satisfy
(16)$$\theta_{ijk}^{lower}\leq \frac{N_{ijk}+\alpha_{ij}\theta_{ijk}^{prior}}{N_{ij}+\alpha_{ij}}\leq \theta_{ijk}^{upper}$$
which implies
(17)$$\alpha_{ij}\geq \frac{N_{ij}\theta_{ijk}^{lower}-N_{ijk}}{\theta_{ijk}^{prior}-\theta_{ijk}^{lower}}$$
(18)$$\alpha_{ij}\geq \frac{N_{ij}\theta_{ijk}^{upper}-N_{ijk}}{\theta_{ijk}^{prior}-\theta_{ijk}^{upper}}.$$

(2) To satisfy the intra-distribution constraint, the CaMAP estimation should satisfy
(19)$$\frac{N_{ijk_{1}}+\alpha_{ij}\theta_{ijk_{1}}^{prior}}{N_{ij}+\alpha_{ij}}\leq \frac{N_{ijk_{2}}+\alpha_{ij}\theta_{ijk_{2}}^{prior}}{N_{ij}+\alpha_{ij}}$$
which implies
(20)$$\alpha_{ij}\geq \frac{N_{ijk_{2}}-N_{ijk_{1}}}{\theta_{ijk_{1}}^{prior}-\theta_{ijk_{2}}^{prior}}$$

(3) To satisfy the cross-distribution constraint, the CaMAP estimation should satisfy
(21)$$\frac{N_{ij_{1}k}+\alpha_{ij_{1}}\theta_{ij_{1}k}^{prior}}{N_{ij_{1}}+\alpha_{ij_{1}}}\leq \frac{N_{ij_{2}k}+\alpha_{ij_{2}}\theta_{ij_{2}k}^{prior}}{N_{ij_{2}}+\alpha_{ij_{2}}}$$
where $\alpha_{ij_{1}}$ and $\alpha_{ij_{2}}$ represent the ESS values of the distributions under the cross-distribution constraint. Thus, we have
(22)$$\alpha_{ij_{1}}\alpha_{ij_{2}}\left(\theta_{ij_{1}k}^{prior}-\theta_{ij_{2}k}^{prior}\right)+\alpha_{ij_{1}}\left(N_{ij_{2}}\theta_{ij_{1}k}^{prior}-N_{ij_{2}k}\right)+\alpha_{ij_{2}}\left(N_{ij_{1}k}-N_{ij_{1}}\theta_{ij_{2}k}^{prior}\right)$$

In this paper, we set $\alpha_{ij_{1}}=\alpha_{ij_{2}}$ and thus we have
(23)$$\alpha_{ij_{1}}^{2}\left(\theta_{ij_{1}k}^{prior}-\theta_{ij_{2}k}^{prior}\right)+\alpha_{ij_{1}}\left(N_{ij_{1}k}-N_{ij_{2}k}+N_{ij_{2}}\theta_{ij_{1}k}^{prior}-N_{ij_{1}}\theta_{ij_{2}k}^{prior}\right)+N_{ij_{2}}N_{ij_{1}k}-N_{ij_{1}}N_{ij_{2}k}\leq 0$$
From the above inequality, constraints on the ESS values $\alpha_{ij_{1}}$ and $\alpha_{ij_{2}}$ could be derived.

Furthermore, in this paper, for each node, we define two classes of ESSs: “*global*” and “*local*” ESS. “*Global*” ESS refers to the equivalent sample size imposed on all parameter distributions of the given node, such as node i, while “*local*” ESS refers to the equivalent sample size working on parameter distribution that refers to a specific parent configuration state. For example, in Figure 1, for node i, $\alpha_{i}$ is the “*global*” ESS, while $\left(\alpha_{i1},\;\ldots ,\alpha_{iq_{i}}\right)$ are the “*local*” ESSs.

In general, with the elicited prior, observational data and parameter constraints, for node i, the optimal ESSs could be determined by the following procedure:

(1) First, from the elicited prior and observational data, the optimal “*global*” *ESS α_i_* could be determined by cross-validation [29]. In the cross-validation, each candidate ESS (In this paper, the candidate ESS varies from 1 to 50) is evaluated based on the likelihood of posteriori estimation in Equation (11).

(2) Then, based on the parameter constraints, we can derive the constraints on each “*local*” ESS $\alpha_{ij}$.

(3) Finally, for “*local*” ESS $\alpha_{ij}$, (I) If there is no constraint imposed on $\alpha_{ij}$, then we set $\alpha_{ij}=\alpha_{i}$. (II) If there are constraints imposed on $\alpha_{ij}$ and meanwhile the “*global*” ESS $\alpha_{i}$ satisfies the constraints, then, we set $\alpha_{ij}=\alpha_{i}$; if not, $\alpha_{ij}$ is determined by further cross-validation using data, prior and ESS constraints. Note that in the process of validation, the initial candidate ESS value of $\alpha_{ij}$ is set to be the lower bound value of the range defined by its constraints.

The pseudo-code of the proposed CaMAP algorithm could be summarized as following Algorithm 1:
**Algorithm 1** Constrained adjusted Maximum a Posteriori (CaMAP) algorithm
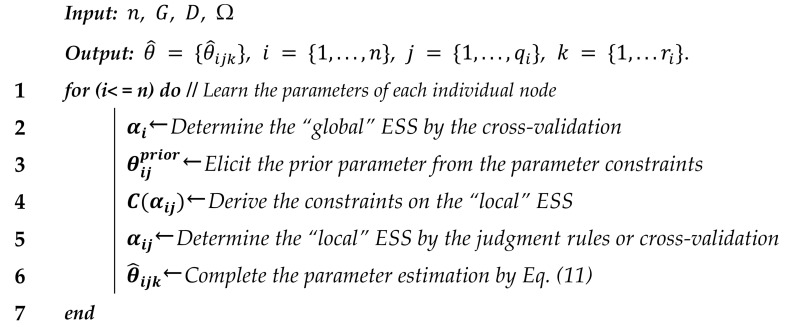


### 3.3. Numerical Illustration of CaMAP Method

To illustrate the principle of the proposed method, we demonstrate the parameter learning of the BN shown in Figure 2, which is extracted from the brain tumor BN [23]. Nodes in the network have meanings as below. Specifically, the network indicates that the presence of brain tumor and the increased level of serum calcium may cause coma.

*C* → *Coma**BT* → *Brain Tumour**IS* → *Increased level of Serum calcium*

(1) First, we assume that a small data set of 20 patients is available. From the data, the following counting are observed:$$\begin{array}{c} N\left(C=0,BT=0,IS=0\right)=0,\;N\left(C=0,BT=0,IS=1\right)=1\\ N\left(C=0,BT=1,IS=0\right)=3,\;N\left(C=0,BT=1,IS=1\right)=9\\ N\left(C=1,BT=0,IS=0\right)=3,\;N\left(C=1,BT=0,IS=1\right)=0\\ N\left(C=1,BT=1,IS=0\right)=4,\;N\left(C=1,BT=1,IS=1\right)=0. \end{array}$$

Furthermore, we acquire the following medical knowledge from the medical experts: a brain tumor as well as an increased level of serum calcium are likely to cause the patient to fall into a coma in due course. From this medical knowledge, we generate the following parameter constraints:$$\begin{array}{c} P\left(C=1|BT=0,IS=1\right)\geq P\left(C=1|BT=0,IS=0\right)\\ P\left(C=1|BT=1,IS=0\right)\geq P\left(C=1|BT=0,IS=0\right)\\ P\left(C=1|BT=1,IS=1\right)\geq P\left(C=1|BT=0,IS=0\right)\\ P\left(C=1|BT=1,IS=1\right)\geq P\left(C=1|BT=0,IS=1\right)\\ P\left(C=1|BT=1,IS=1\right)\geq P\left(C=1|BT=1,IS=0\right). \end{array}$$

(2) Then, based on the parameter constraints, we elicit the following priors using the proposed prior elicitation algorithm (Section 3.1):$$\begin{array}{c} P^{\prime }\left(C=0|BT=0,IS=0\right)=0.99,\;P^{\prime }\left(C=0|BT=0,IS=1\right)=0.56\\ P^{\prime }\left(C=0|BT=1,IS=0\right)=0.60,\;P^{\prime }\left(C=0|BT=1,IS=1\right)=0.05\\ P^{\prime }\left(C=1|BT=0,IS=0\right)=0.01,\;P^{\prime }\left(C=1|BT=0,IS=1\right)=0.44\\ P^{\prime }\left(C=1|BT=1,IS=0\right)=0.40,\;P^{\prime }\left(C=1|BT=1,IS=1\right)=0.95 \end{array}$$

(3) Furthermore, from the parameter constraints, we derive the constraints on the “*local*” ESSs:$$\begin{array}{c} \alpha \left(BT=0,IS=0\right)\geq 5.49,\;\alpha \left(BT=0,IS=1\right)\geq 5.92\\ \alpha \left(BT=1,IS=0\right)\geq 9.01,\;\alpha \left(BT=1,IS=1\right)\geq 9.01 \end{array}$$

(4) Next, for node C, the optimal “*global*” ESS is cross-validated to be 3. As the “*global*” ESS does not satisfy any of the ESS constraints, the “*local*” ESSs would not be equal to the “*global*” ESS and should be further validated. Based on the prior, data and ESS constraints, the optimal “*local*” ESSs are cross-validated to be as follows:$$\begin{array}{c} \alpha \left(BT=0,IS=0\right)=50,\;\alpha \left(BT=0,IS=1\right)=6\\ \alpha \left(BT=1,IS=0\right)=50,\;\alpha \left(BT=1,IS=1\right)=50 \end{array}$$

(5) Finally, with the elicited priors and optimal ESSs, the CaMAP estimation are computed as follows:$$\begin{array}{c} P\left(C=0|BT=0,IS=0\right)=\frac{0+50\times 0.99}{3+50}=0.93\\ P\left(C=0|BT=0,IS=1\right)=\frac{1+6\times 0.56}{1+6}=0.62\\ P\left(C=0|BT=1,IS=0\right)=\frac{3+50\times 0.60}{7+50}=0.58\\ P\left(C=0|BT=1,IS=1\right)=\frac{9+50\times 0.05}{9+50}=0.19\\ P\left(C=1|BT=0,IS=0\right)=\frac{3+50\times 0.01}{3+50}=0.07\\ P\left(C=1|BT=0,IS=1\right)=\frac{0+6\times 0.44}{1+6}=0.38\\ P\left(C=1|BT=1,IS=0\right)=\frac{4+50\times 0.40}{7+50}=0.42\\ P\left(C=1|BT=1,IS=1\right)=\frac{0+50\times 0.95}{9+50}=0.81 \end{array}$$

## 4. The Experiments

We conducted experiments to investigate the performance of the proposed CaMAP method in terms of learning accuracy, under different sample sizes and constraint sizes. In the experiments, we used the networks from [16,17], shown in Figure 3, Figure 4, Figure 5, Figure 6 and Figure 7. The true parameter distributions in these networks show different uniformities, varying from strongly skewed to strongly uniform distributions. As the true parameters were set or known in advance, the learnt parameters were evaluated by the Kullback–Leibler (KL) divergence [30], which indicates the divergence between the learnt parameters or estimated distribution and the true parameters or underlying distribution. The proposed method was evaluated against the following learning algorithms: ME [31], ML [32], MAP [13], CME [26,33], and CML [24,34] (The code of all the six tested algorithms can be found at https://github.com/ZHIGAO-GUO/CaMAP (accessed on 26 September 2021)). The full names of the tested algorithms are listed as follows:ME: maximum entropyML: maximum likelihoodMAP: maximum a posterioriCME: constrained maximum entropyCML: constrained maximum likelihoodCaMAP: constrained adjusted maximum a posteriori


Notice that, (I) in the MAP method, we used a uniform (or flat) prior, which means, $\theta_{ijk}^{prior}$ in Equation (11) was set to be $1/r_{i}$ and ESS value is 1, and (II) in the CaMAP method, we set the maximum candidate ESS to be 50, which is a sufficient number for all networks.

### 4.1. Learning with Different Sample Sizes

First, we examined the learning performance of all algorithms under different sample sizes. Our experiments were carried out under the following settings: (1) The sample sizes were set to be 10, 20, 30, 40, and 50, respectively. (2) The parameter constraints were randomly generated from the true parameters of the tested networks, with the maximum number of constraints for each node at 3. Specifically, the parameter constraints are generated using the following rules: (1) Range constraints are generated as $\left[\theta_{ijk}^{lower},\theta_{ijk}^{upper}\right]$, where $\theta_{ijk}^{lower}$ is equal to be $max\left(0,\theta_{ijk}^{*}-\tau_{1}\right)$ and $\theta_{ijk}^{upper}$ is equal to be $min\left(1,\theta_{ijk}^{*}+\tau_{2}\right)$, where $\theta_{ijk}^{*}$ represents the true parameter, and $\tau_{1}$ and $\tau_{2}$ are two random values around 0.2. (2) Inequality constraints are generated as $\theta_{ij_{1}k_{1}}\geq \theta_{ij_{2}k_{2}}$ if $\left(\theta_{ij_{1}k_{1}}-\theta_{ij_{2}k_{2}}\right)\geq 0.2$. Therefore, when $j_{1}=j_{2}$ and $k_{1}\neq k_{2}$, the constraint becomes the intra-distribution constraint, while the constraint becomes the cross-distribution constraint when $j_{1}\neq j_{2}$ and $k_{1}=k_{2}$,.

We performed 100 repeated experiments. The average KL divergence values of different algorithms on different networks under different sample sizes are summarized in Table 1 with the best results highlighted in bold.

From the experimental results, we draw the following conclusions: (1) With increasing data, the performance of all algorithms improved by different levels. (2) In almost all cases, CaMAP outperformed the other learning algorithms. However, when the available data are extremely insufficient, e.g., 10, the CaMAP was inferior to the MAP method. The explanation might be that the insufficiency of data impacts the cross-validation of ESS values. Therefore, the optimal ESS turns out to be extreme, either small or large, and fails to balance data and prior (see the 2nd future study in Discussion and Conclusions section).

### 4.2. Learning with Different Constraint Sizes

Next, we further explored the learning performance of different learning algorithms under different constraint sizes. The experiments were conducted under the following settings: (1) The data set size for all the tested networks was set to be 20, which is a small number for all networks. (2) Parameter constraints were generated from the true parameters of the networks and the maximum number of constraints for each node was set to be 3. The parameters were learnt from a fixed data set but an increasing number of parameter constraints that were randomly chosen from all generated constraints. The constraint sparsity varied from 0% to 100%. For each setting, we performed 100 repeated experiments. The average KL divergence values of different algorithms on different networks under different constraint sizes are summarized in Table 2.

From the experimental results, we draw the following conclusions: (1) For the algorithms that did not use constraints, such as ML, ME, and MAP, changing the constraint size did not impact their performance. However, for the algorithms that have been incorporated constraints, such as CML, CME, and CaMAP, an increase in constraints affected their performances to a certain degree depending on the number of incorporated constraints. (2) In most cases, CaMAP outperformed the other parameter learning algorithms, except for MAP, when no parameter constraints were incorporated into the learning. In fact, when no parameter constraints were available, CaMAP method was slightly inferior to the MAP estimation with uniform prior. The explanation might be as follows: when the parameter constraints are not available, constraints on ESS values could not be deduced. Therefore, ESS values in CaMAP estimation are the same at those in MAP estimation. Then, the difference between the CaMAP and MAP estimation lies in the prior, $\theta_{i}^{prior}$. However, unlike uniform prior in MAP estimation, prior in the CaMAP method is elicited using a sampling method. For the sampling methods, it is hard to achieve completely uniform sampling unless the sampling size is very large (see the 1^st^ future study in the Discussion and Conclusions section).

## 5. Discussion and Conclusions

For MAP estimation in BN parameter learning, informative prior distribution and reasonable ESS values are two crucial factors that impact the learning performance. Empirically, a uniform prior is preferred and ESS is further cross-validated according to the uniform prior. However, when the underlying parameter distribution is non-uniform or skewed, MAP estimation with a uniform prior does not fit the underling parameter distribution well, and, in that case, an informative prior is required. In fact, reliable qualitative domain knowledge has been proved to be useful and can be used for eliciting informative priors and selecting the reasonable ESS. In this paper, we proposed a CaMAP estimation method. The proposed method automatically elicits the prior distribution from the parameter constraints that are transformed from the domain knowledge. Besides, constraints on ESS values are derived from the parameter constraints. Then, the optimal ESS, including “*global*” and “*local*” ESS, are further chosen from the ranges derived from the ESS constraints by cross-validation. Our experiments demonstrated that the proposed method outperformed most of the mainstream parameter learning algorithms. In future study:

(1) A more effective prior elicitation approach is desired. Compared to the sampling-based methods, geometric constraint-solving methods would be more robust and could elicit more informative priors.

(2) A more reasonable ESS selection method is preferred. For the cross-validation method, when the available data are extremely insufficient or less informative, the optimal ESS tends to maximize the likelihood of data and makes the CaMAP estimation fail to approach the underling parameter distribution. In fact, data bootstrapping guided by the parameter constraints may extend the data and make the data more informative and thus improve the ESS selection.

## Figures and Tables

**Figure 1 entropy-23-01283-f001:**
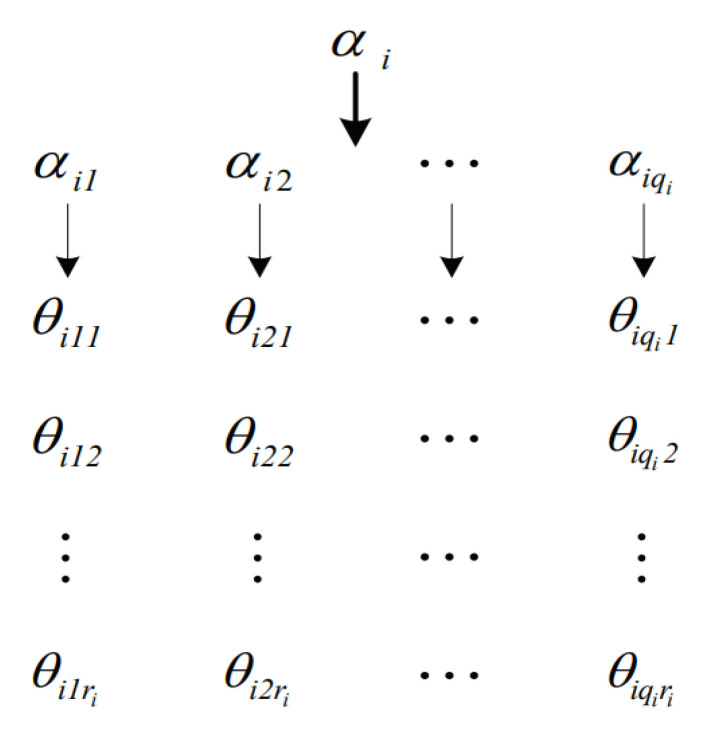
Illustration of “*global*” and “*local*” ESS.

**Figure 2 entropy-23-01283-f002:**
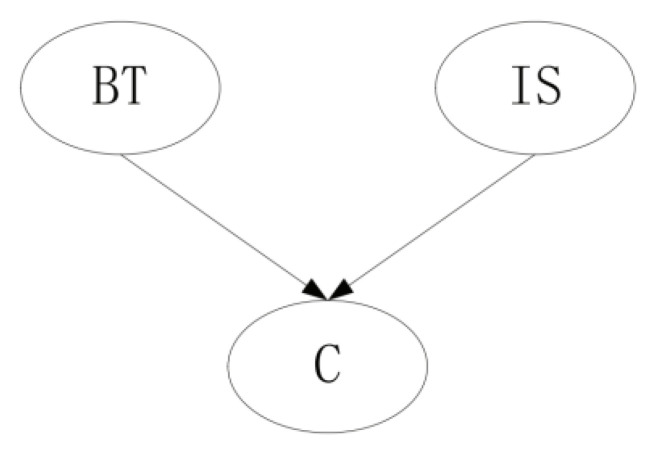
Brain tumor BN.

**Figure 3 entropy-23-01283-f003:**
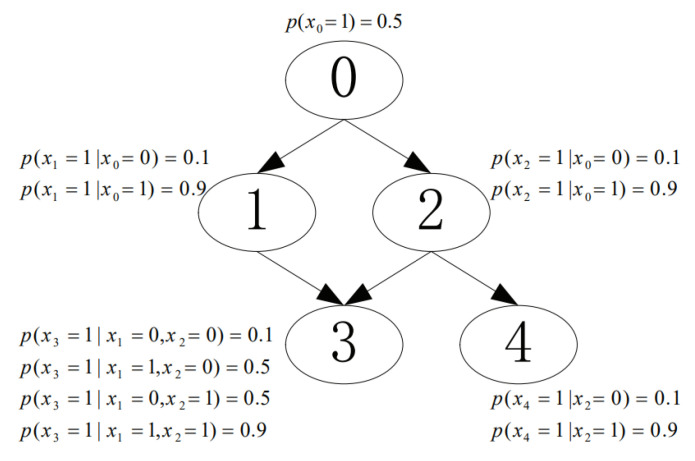
Strongly skewed distribution.

**Figure 4 entropy-23-01283-f004:**
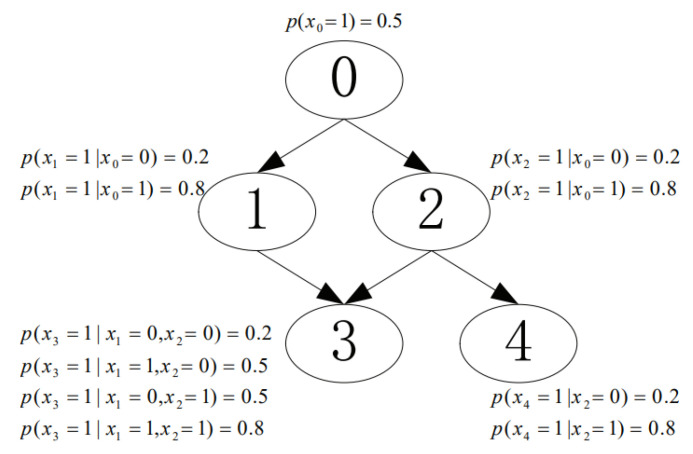
Skewed distribution.

**Figure 5 entropy-23-01283-f005:**
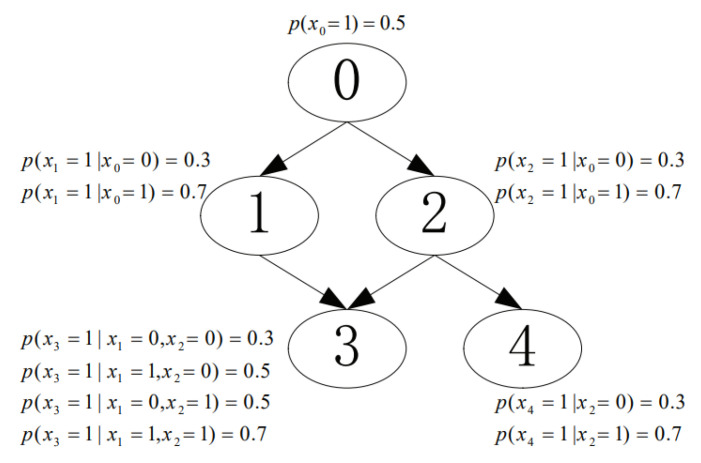
Uniform distribution.

**Figure 6 entropy-23-01283-f006:**
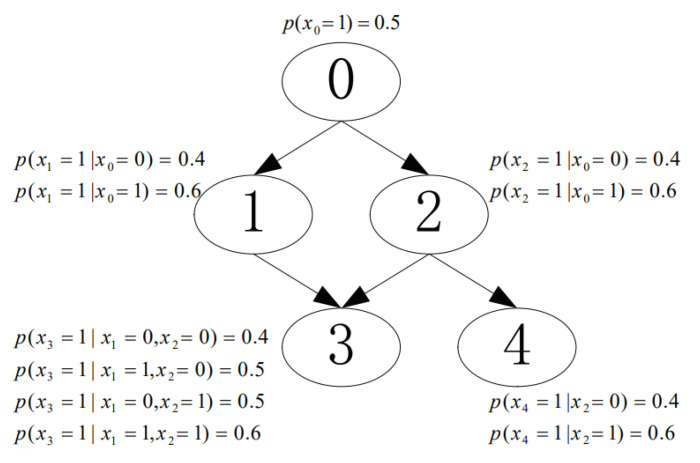
Strongly uniform distribution.

**Figure 7 entropy-23-01283-f007:**
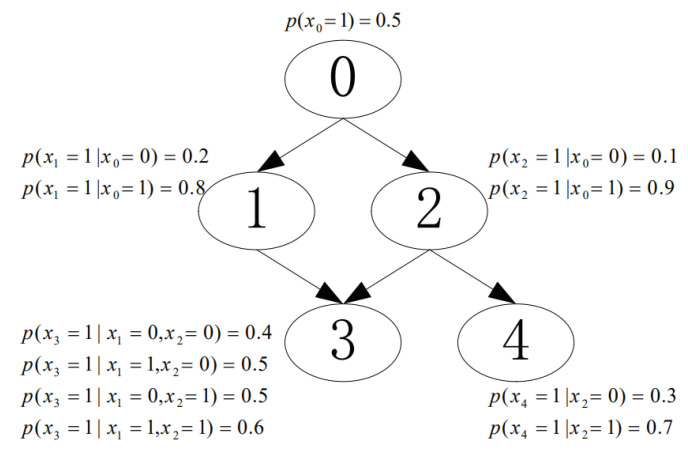
Combined skewed and uniform distribution.

**Table 1 entropy-23-01283-t001:** Parameter learning under different sample sizes.

	ML	CML	ME	CME	MAP	CaMAP
(a) Network—strongly skewed distribution						
10	2.455	0.946	0.196	0.098	**0.083**	0.108
20	1.234	0.486	0.131	0.075	0.066	**0.063**
30	0.486	0.211	0.070	0.046	0.050	**0.038**
40	0.291	0.147	0.053	0.036	0.040	**0.029**
50	0.192	0.098	0.044	0.033	0.034	**0.024**
(b) Network—skewed distribution						
10	2.277	0.884	0.182	0.090	**0.077**	0.104
20	1.170	0.481	0.122	0.068	**0.062**	0.064
30	0.589	0.257	0.085	0.055	0.055	**0.042**
40	0.302	0.139	0.060	0.042	0.046	**0.030**
50	0.154	0.066	0.044	0.034	0.037	**0.025**
(c) Network—uniform distribution						
10	2.350	1.060	0.195	0.095	**0.072**	0.103
20	1.036	0.452	0.118	0.070	**0.066**	0.069
30	0.515	0.229	0.080	0.053	0.060	**0.049**
40	0.238	0.101	0.053	0.039	0.044	**0.029**
50	0.150	0.069	0.040	0.030	0.037	**0.023**
(d) Network—strongly uniform distribution						
10	2.102	0.899	0.182	0.091	**0.070**	0.105
20	1.202	0.528	0.122	0.064	0.063	**0.060**
30	0.470	0.214	0.075	0.047	0.054	**0.040**
40	0.353	0.151	0.062	0.041	0.045	**0.030**
50	0.186	0.057	0.043	0.031	0.034	**0.021**
(e) Network—combined skewed and uniform distribution						
10	2.460	1.015	0.201	0.097	**0.074**	0.102
20	1.103	0.433	0.121	0.069	0.066	**0.058**
30	0.631	0.228	0.089	0.055	0.053	**0.042**
40	0.290	0.126	0.061	0.043	0.047	**0.028**
50	0.206	0.097	0.051	0.038	0.039	**0.025**

**Table 2 entropy-23-01283-t002:** Parameter learning under different constraint sizes.

	ML	CML	ME	CME	MAP	CaMAP
(a) Network—strongly skewed distribution						
0%	1.321	1.023	0.133	0.097	**0.080**	0.082
25%	1.321	0.691	0.133	0.092	0.080	**0.057**
50%	1.321	0.382	0.133	0.083	0.080	**0.045**
75%	1.321	0.168	0.133	0.069	0.080	**0.022**
100%	1.321	0.063	0.133	0.055	0.080	**0.005**
(b) Network—skewed distribution						
0%	1.313	1.003	0.131	0.093	**0.077**	0.080
25%	1.313	0.554	0.131	0.090	0.077	**0.052**
50%	1.313	0.345	0.131	0.082	0.077	**0.041**
75%	1.313	0.098	0.131	0.072	0.077	**0.017**
100%	1.313	0.065	0.131	0.054	0.077	**0.005**
(c) Network—uniform distribution						
0%	1.184	0.925	0.127	0.094	**0.073**	0.075
25%	1.184	0.505	0.127	0.091	0.073	**0.052**
50%	1.184	0.241	0.127	0.083	0.073	**0.037**
75%	1.184	0.118	0.127	0.071	0.073	**0.017**
100%	1.184	0.058	0.127	0.055	0.073	**0.007**
(d) Network—strongly uniform distribution						
0%	1.303	0.999	0.126	0.093	**0.072**	0.073
25%	1.303	0.724	0.126	0.089	0.072	**0.052**
50%	1.303	0.474	0.126	0.078	0.072	**0.039**
75%	1.303	0.196	0.126	0.067	0.072	**0.023**
100%	1.303	0.072	0.126	0.049	0.072	**0.007**
(e) Network—combined skewed and uniform distribution						
0%	1.170	0.900	0.121	0.088	**0.076**	0.080
25%	1.170	0.512	0.121	0.084	0.076	**0.050**
50%	1.170	0.296	0.121	0.077	0.076	**0.025**
75%	1.170	0.153	0.121	0.068	0.076	**0.014**
100%	1.170	0.050	0.121	0.050	0.076	**0.005**

## Data Availability

Data used in the experiments are synthetically generated from the networks (refer to Figure 3, Figure 4, Figure 5, Figure 6 and Figure 7) and could be generated by the open-source code provided in the paper.

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
