# Peer review of "Constrained Adjusted Maximum a Posteriori Estimation of Bayesian Network Parameters"

_entropy, 2021, doi:10.3390/e23101283_

Round 1

Reviewer 1 Report

Page 1, Line 28: BN models are not causal models unless in some special instances

Page 1, Line 33: Define what a DAG is

Page 2, Line 41: It should be hyper-parameters of the BN model because BN model has parameters and each parameter can have a prior distribution that has some parameters. The parameters of the prior distribution are called hyper-parameters.

Page 2, Line 42: Dirichlet priors are used only for BN nodes that has multinomial distributions. That is, such nodes represent discrete variables. This paper is all about discrete variable BNs

Page 2, Kline 44-49: Carelessly written. Define each notation precisely. \alpha_{ij} is the sum over all k,  \alpha_{ijk} and so on. Follow the literature and use standard notation

Page 2, Line 55: why do authors say a numerical fraction uniform (or non-uniform). Use the correct terminology to avoid confusion. For example, a distribution can be uniform but not its components/contents. Avoid such errors elsewhere too.

Page 3, Line 86: Equation (1) is true if the joint distribution of the set of the variables facterizes over the DAG that has been selected. Avoid the decomposability since it is not defined here.

Page 4, Line 132: Prior distribution for \theta_{ij} is taken as Dirichlet since it is the conjugate distribution for the multinomial distribution. Note that.

Page 4, Line 136: It is better to use the phrase “hypothetical sample size” rather that equivalent sample size, because prior information is hypothetical

Page 4, Line 141: Notation in the equation (9) is not consistent with that in the Line 133

Page 4, Line 142: Why do the authors call a parameter prior distribution?

Page 5, Line 147: More suitable name is Constrained adjusted MAP estiation since it is really the case that MAP estimator is adjusted according to the constraints.

Page 5, Line 174: Present your case as an example with specific I, j and k, and using the word “if”. In here the language used is not showing that it is an example but general assumption.

Page 6, Line 181: Is it optimal or just correct? Since ESS is a hypothetical count, what is ypur argument for selecting a number for it so that it only satisfies computational need? Suppose you need to you a large number which is unrealistic for domain expert. Note that in Bayesian analysis, these hypothetical counts represent extent to which the domain expert has seen the event of interest in the past.

Page 6, Line 191: Is a bracket missing in the numerator in the right-hand side? There are many places where there are missing brackets, etc. in expressions  

Page 7, Line 225: The example given is too simple. It has only one independence assumption that is marginal independence of BT and IS. A little bigger network is preferred.

Page 9, Line 274: The experiment seems to be fine but I did not take much time to review it since the paper needs major revisions as far as description of the topic is concern.

Follow the book "Probabilistic Networks and Expert Systems" by Cowell, Dawid, Lauritzen and Spiegelhalter (1999) Springer for notations and interpretations 

Reviewer 2 Report

page 1, line 30: typo 2 - must be "widely"

page 2, lines 45-48: N_{ijk} and N_{ij} should be defined with more mathematical precision

page 2, lines 51-52: BDu, Deu and BDs not explained or defined

page 3, line 87: typo, should be "node"

page 3, lines 94-95: what is "sufficient" and "insufficient" ? That's too imprecise

page 3, lines 99: what is a "convex" constraint ? Please explain !

page 4, (5) please provide an example

page 5, (10): please derive the logic of (10)

page 5, (11): what is "C" ? There is no definition or explanation

page 6, (12): what is "s.t." ? Please define or explain

page 6, lines 160-161: please explain in a special appendix

page 7, Figure 1: the logic of this figure remains unclear. Please give a better explanation or dele that figure

page 7, lines 214-224: Please provide pseudocode in an extra appendix

page 8, lines 240-242: this medical knowledge contradicts the "data" in lines 236-239 ! What should the reader deduce from such a contradiction ?

page 8, line 266: the conditional P(...) = 0.93 contradicts the data !

page 9, line 290: what's the meaning of "(1) ?

page 14, line 368: what's the meaning of "cross-validation" when there is only *one* data set.

Please provide the new algorithms as pseudo-code in special appendices

Round 2

Reviewer 1 Report

First of all, I am not convinced why it is necessary to have CaMAP estimate should satisfy the prior constraints. If it should satisfy then prior information controls or dominate the real observations. Suppose expert opinion is not correct, then what are the consequences of your estimation method. Maybe, you need to clarify this to a good extent.

Still the paper need to be improved as far as notations, definitions, concepts, etc. are concern. You may refer Cowell et al. (1999)

Probabilistic Networks and Expert Systems. Springer. https://www.springer.com/gp/book/9780387987675

Paper should have its contents mathematically precise. Still the paper lacks it. And my comments (below) are on that before I can make those on methodical contents and matters.

Line 29: desired causal assumptions

Line 30: statistical prediction and classification problems

Line 35: directed edges

Line 36: conditional dependence relationships among the variables. [Also add this sentence:] The graph has no directed cycles.

Line 47: …..and its set of parents has the jth state of its configurations

Line 48:  the sum of $N_{ijk} $ over all k

Line 49: For all k, $\apha_{ijk}$ are the hyper-parameter values of the Dirichlet prior distribution of the BN parameter $\theta_{ijk}$

Line 56: purely data-driven estimation that gives the estimate $\N_{ijk} / N_{ij}$ for the $\theta_{ijk}$.......

Line 58: [adjust the sentence as for Line 56]

Line 128: [it is standard to write ]   $\hat{\theta}=( \hat{\theta}_1,…,\hat{\theta}_n )$,

$\hat{\theta}_i=( \hat{\theta}_{i1},…,\hat{\theta}_{iq_i} )$,

$\hat{\theta}_{ij}=( \hat{\theta}_{ij1},…,\hat{\theta}_{ijr_i} )$,

Here; n is the number of variables in the network, $q_i$ is the number of configurations of the parent set of the variable $X_i$  and $r_i$ is the number of configurations of the variable $X_i.$

Line 158 and 159  [Instead of $Dir(\alpha_{ij})$, it should be written as, for example, $Dir(\alpha_{ij}*\theta_{ij}^{prior})$ where * denotes vector component-wise product, in order to have consistent notation. ]

Line 160: ….. is the prior hyper-parameter vector of the prior distribution that …..

Line 166: [Do not call $\theta_{ij}^{prior}$] the prior distribution but call it prior parameter. Do so elsewhere too.

Line 183 [Footnote is incomplete]

Line 217 [is the inequality sign right?]

Line 252 [Indicate the value ranges if i, j and k]

Line 280-284 [ Aren’t they conditional probabilities]

Line 308: [Why have you selected KL-divergence as to indicate the "difference" between true parameter and the estimated parameter. does it work good? Are there any other measures for the task]

Improve the presentations in Experiment and, Discussion and Conclusion sections 

Reviewer 2 Report

is OK now

Author Response

This manuscript is a resubmission of an earlier submission. The following is a list of the peer review reports and author responses from that submission.